# Immunomodulating Effects of Fungal Beta-Glucans: From Traditional Use to Medicine

**DOI:** 10.3390/nu13041333

**Published:** 2021-04-17

**Authors:** Hidde P. van Steenwijk, Aalt Bast, Alie de Boer

**Affiliations:** 1Campus Venlo, Food Claims Centre Venlo, Faculty of Science and Engineering, Maastricht University, 5911 BV Venlo, The Netherlands; a.deboer@maastrichtuniversity.nl; 2Campus Venlo, University College Venlo, Maastricht University, 5911 BV Venlo, The Netherlands; a.bast@maastrichtuniversity.nl; 3Department of Pharmacology & Toxicology, Medicine and Life Sciences, Faculty of Health, Maastricht University, 5911 BV Venlo, The Netherlands

**Keywords:** mushrooms, immunomodulation, nutrition, dietary supplement, health claims, medicine, immunity

## Abstract

The importance of a well-functioning and balanced immune system has become more apparent in recent decades. Various elements have however not yet been uncovered as shown, for example, in the uncertainty on immune system responses to COVID-19. Fungal beta-glucans are bioactive molecules with immunomodulating properties. Insights into the effects and function of beta-glucans, which have been used in traditional Chinese medicine for centuries, advances with the help of modern immunological and biotechnological methods. However, it is still unclear into which area beta-glucans fit best: supplements or medicine? This review has highlighted the potential application of fungal beta-glucans in nutrition and medicine, reviewing their formulation, efficacy, safety profile, and immunomodulating effects. The current status of dietary fungal glucans with respect to the European scientific requirements for health claims related to the immune system and defense against pathogens has been reviewed. Comparing the evidence base of the putative health effects of fungal beta-glucan supplements with the published guidance documents by EFSA on substantiating immune stimulation and pathogen defense by food products shows that fungal beta-glucans could play a role in supporting and maintaining health and, thus, can be seen as a good health-promoting substance from food, which could mean that this effect may also be claimed if approved. In addition to these developments related to food uses of beta-glucan-containing supplements, beta-glucans could also hold a novel position in Western medicine as the concept of trained immunity is relatively new and has not been investigated to a large extent. These innovative concepts, together with the emerging success of modern immunological and biotechnological methods, suggest that fungal glucans may play a promising role in both perspectives, and that there are possibilities for traditional medicine to provide an immunological application in both medicine and nutrition.

## 1. Introduction

Many chronic diseases can be explained by an underlying chronic inflammation; more appealing to the imagination is the recent COVID-19 pandemic, which has presented the modern world with a challenge that global health care has not faced in more than a century since the Spanish flu pandemic in 1918 [1]. A characteristic feature of an infection with COVID-19 is a pro-inflammatory status characterized by high levels of different cytokines, including interleukin (IL)-1β, IL-1Rα, IL-2, IL-10, fibroblast growth factor (FGF), granulocyte-macrophage colony stimulating factor (GM-CSF), granulocyte-colony stimulating factor (G-CSF), interferon-γ-inducible protein (IP10), monocyte chemoattractant protein (MCP1), macrophage inflammatory protein 1 alpha (MIP1A), platelet-derived growth factor (PDGF), tumor necrosis factor (TNF-α), and vascular endothelial growth factor (VEGF) [2]. These changes in cytokine levels are associated with various changes in cellular components of the immune response [3]. It becomes more evident that there is a close interaction between the virus and an individual’s immune system, resulting in different clinical manifestations of the disease [4]. Moreover, with the aid of modern immunological and biotechnological methods, the importance of a well-functioning and balanced immune system in maintaining overall health has become more apparent in recent decades [5]. In anticipation of the global immunization of the population through vaccines developed for the adaptive immune system, innate immune-based strategies for therapeutic purposes are also being investigated [2]. The innate immune system constitutes the host’s first line of defense during infection and therefore plays a critical role in the early recognition and subsequent activation of a pro-inflammatory response to invading pathogens (for review, see [6]). Nonspecific immunostimulants (NSIs) are natural, synthetic, or recombinant molecules that stimulate the innate immune system by inducing activation or increasing activity of any of its components. In contrast to specific immunostimulants such as vaccines, NSIs act irrespective of antigenic specificity to augment immune response of other antigen or stimulate components of the immune system without antigenic specificity. Despite the tremendous advances in this field of immunology over the years, many areas of uncertainty remain. For example, one question that remains is what is the precise mechanism of action of cell activation, immunomodulation, and tumor reduction for NSIs used in cancer therapy (e.g., mifamurtide, BCG vaccine). In addition to increasing our understanding of these drugs, there is an increasing interest in the development of NSIs that can be used in infectious and inflammatory diseases [6]. Recently, immunomodulators used in traditional Chinese medicine (TCM) for centuries, such as the shiitake and the pearl oyster mushroom, have gained interest for these new developments [7]. Many of the traditionally used substances, however, are substantiated by only limited scientific studies. An exception to this is the fungal beta-glucans which, with more than 20,000 published studies, are the most studied mushroom-derived molecules with potential immunomodulating properties [8,9].

### Structure, Chemical Properties, and Natural Sources of Beta-Glucans

Beta-glucans are groups of polysaccharides or dietary fibers composed of D-glucose monomers, linked by (1 → 3), (1 → 4) or (1 → 6) glycosidic bonds. Beta-glucans are naturally found in the cell wall of bacteria, fungi, algae, and cereals such as oat and barley [10]. The different sources of beta-glucans, however, also differ in the linkages between the D-glucose monomers. The beta-glucans present in cereals include a mixture of (1 → 3) and (1 → 4) glycosidic bonds. Beta-glucans in mushrooms mostly contain a linear (1 → 3) backbone with (1 → 6)-linked glucose branches attached. Furthermore, beta-glucans found in yeasts, seaweeds, and bacteria display different structural forms and branching, of which curdlan, extracted from *Agrobacterium*, is the simplest structure; it is only composed of unbranched (1 → 3) glycosidic bonds [11]. These differences in the shape, structure, and molecular weight of beta-glucans determine their biological activity. For cereal beta-glucans (dietary fibers), mainly physicochemical properties are reported, including reactive oxygen species (ROS) scavenging activity, and their ability to lower serum cholesterol and improve gut microbiome [11]. These properties have been attributed to the mixture of only the (1 → 3) and (1 → 4) glycosidic bonds, making them resistant to absorption and digestion in the small intestine of humans [12,13]. The effects of beta-glucans containing (1 → 6) branching, such as fungal or bacterial glucans, are related to the activation/inactivation of specific receptors such as dectin-1 (mostly insoluble beta-glucans), complement receptor 3 (CR3), or toll-like receptor 2 (TLR2) (mainly water-soluble glucans) [11,14]. Individual fungi contain specific beta-glucans, which differ from each other by the amount of (1 → 6) linked side chains (Figure 1). Moreover, the content and proportions of beta-glucans in fungi is mainly determined by their genetic profile and differs between species and even cultivars. Upon ingestion, fungal glucans affect the mucosal immune system in the gastrointestinal tract. Similar to antigens, the uptake of beta-glucans occurs via microfold cells (M cells) localized within Peyer’s patches in the small intestine. M cells subsequently present the antigen or beta-glucan at their basal surfaces to immune cells, such as macrophages and dendritic cells. Here, beta-glucan particles bind with macrophages with the help of dectin-1, the primary receptor for most insoluble beta-glucans. Subsequently, dectin-1 induces the secretion of pro-inflammatory cytokines via nuclear factor kappa-B (NF-κB) and various interconnected inflammatory and immunoregulatory processes such as chemokinesis and chemotaxis (for review, see [15]). Given these immunomodulatory effects, the use of fungal glucans as pharmaceutical agents, which act as (adjuvant) immunomodulators, has been authorized in several countries, including the United States of America, Canada, Finland, Sweden, China, Japan, and Korea [16]. Lentinan, isolated from shiitake mushroom, is an example of a pharmaceutically formulated polysaccharide approved as an intravenous immunostimulant in the treatment of multiple cancers in China and Japan [17]. In addition, fungal glucans are widely used in the nutritional field as dietary supplements. Unlike pharmaceutical drugs that are supposed to suppress or stimulate the immune system in patients, dietary supplements are primarily intended as a daily oral dose to support the immune system in healthy people. Pleuran, isolated from pearl oyster mushrooms, is an example of a polysaccharide developed as a dietary supplement to support the immune system and overcome the first signs of exhaustion and fatigue in adults and children [18]. In addition to these formulations, the edible mushrooms shiitake (*Lentinula edodes*) and the pearl oyster mushroom (*Pleurotus ostreatus*) are some of the main dietary sources of beta-glucans (Figure 1) [19].

Modern immunological and biotechnological methods provide us with increasing insight into the effects and function of beta-glucans [8]. However, there are still many unresolved questions, for example, one of the biggest challenges remains the standardization and correct characterization of the molecules themselves. Furthermore, it is still unclear into which area beta-glucans fit best: supplements or medicine? In this review, we shed light on both perspectives to discover which area the beta-glucans, in addition to TCM, could fit, and what evidence is needed for this. In the field of dietary supplements, we highlight the evidence required to use health claims on products according to regulatory authorities in Europe. In the field of medicine, the potential pharmaceutical application of beta-glucans within a novel concept in immunology, namely trained immunity, is discussed.

## 2. Beta-Glucans in Immunomodulating Dietary Supplements

### 2.1. Health Claims in Europe

Supplement manufacturers are interested in using health claims on their product to show the health benefits of consuming these products [20,21,22]. In Europe, the use of voluntary statements related to either the nutritional content (nutrition claims) or health benefits of a food product (health claims) is regulated under the Nutrition and Health Claim Regulation (NHCR). Before such claims can be used on foods, they need to be authorized by the European Commission [23]. Scientific evidence is key in the authorization decision of the European Commission to allow new claims in the EU market [24]. A health claim is defined as any voluntary statement that refers to the relationship between food and health. Four categories of claims are known in the EU: two types of function claims that are based on generally accepted (general function claims, in Art. 13.1) or newly developed scientific evidence (new function claims, Art. 13.5), reduction of disease risk claims (Art. 14.1a) and, finally, claims referring to children’s development and health (Art. 14.1b) [23]. The food business operator needs to submit a scientific dossier along with its request for authoring a newly proposed claim. The European Commission subsequently asks the European Food Safety Authority (EFSA) to evaluate the evidence on the proposed claims [25]. This evaluation involves a critical review of three main criteria: (1) the bioactive substance is sufficiently characterized, (2) the proposed claim is well characterized and should comprise a beneficial physiological effect, and (3) the cause-and-effect relationship between the bioactive substance and the beneficial physiological effect should be established [23,24]. A beneficial effect shown in at least two independently conducted intervention trials increases the chance of receiving a positive opinion of EFSA [20,24]. Since the assessment procedure follows this specific order of evaluation of these criteria, the assessment will be discontinued if the evidence is insufficiently supporting a criterion [26].

### 2.2. Immune Functioning Health Claims

An effective functioning immune system is crucial for maintaining physiological integrity and, thus, for maintaining health. The immune system provides defense against infections caused by pathogenic microorganisms [27,28]. In recent years, food companies have continued to develop innovative foods in this specific field [5]. Despite its positive aim of fostering innovation, the Nutrition and Health Claims Regulation (EC) No. 1924/2006 (NHCR) may present several compliance challenges which might affect innovation in the EU food sector [29,30]. In order to provide stakeholders with greater clarity on which health effects related to immunology could be studied to support health claims, in 2011, a guidance was published by EFSA’s Panel on Dietetic Products, Nutrition and Allergies (NDA Panel) that provides more detailed guidelines for the evaluation of Articles 13.1, 13.5, and 14 health claims in this area [31]. According to the NDA Panel, maintaining a well-functioning immune function can be considered to be a beneficial physiological effect. However, given the multiple roles of the immune system, the specific aspect of immune function to which the claim relates should be noted. This means that changes in multiple biomarkers can indicate a well-functioning immune system. Markers of immune system functioning that are proposed as suitable outcomes for substantiating claims on immune function effects are listed in Table 1 [31]. In addition to a positive influence on these markers, changes should be accompanied by a favorable physiological or clinical outcome, preferably demonstrated in the same study [31].

So far, ten proposed health claims have been considered to be substantiated with sufficient scientific evidence according to the NDA Panel, and have subsequently been authorized by the European Commission. The following six vitamins are reported to play a role in maintaining a well-functioning immune system: A, B6, folate (B9), B12, C, and D [32,33,34]. Meanwhile, a similar assessment was made for four essential trace elements: zinc, copper, iron, and selenium, which are considered by EFSA as necessary for the optimal functioning of the immune system [35,36,37,38,39]. Therefore, these ten micronutrients may be labeled with the health claim ‘contributes to the normal functioning of the immune system’. Moreover, foods containing 200 mg or more vitamin C may be labeled with an additional health claim: ‘Vitamin C contributes to maintain the normal function of the immune system during and after intense physical exercise’ [40]. The evidence for vitamin C’s additional health claim comes from three systematic reviews examining the role of vitamin C supplementation in the prevention, severity, and treatment of the common cold [41,42,43]. The results of the reviews showed that there is some evidence suggesting that individuals who are exposed to short periods of vigorous exercise and/or cold environments benefit from regular vitamin C intake above 200 mg/day based on the duration and severity of the common cold [40]. Next to this second substantiated claim for the effect of vitamin C, the NDA Panel considered that the role of vitamin D in the functioning of the immune system applies to all ages, including children. Therefore, vitamin D containing products may also use an additional Article 14.1(b) claim: ‘Vitamin D contributes to the normal function of the immune system in children’ [40]. This does not mean that other nutrients and foods are not also relevant; so far, however, insufficient scientific evidence has been gathered to demonstrate this [44]. In addition to the 2011 guidelines for claims related to support of the immune system, the guidelines were extended 5 years later to include claims related to stimulation of the immune system and defense against pathogenic microorganisms [31,40]. The scientific evidence to substantiate a claim related to the body’s defense against pathogens can be obtained by studying effects on clinical outcomes related to infections (e.g., incidence, severity, and/or duration of symptoms). As put forward in EFSA’s guideline, the infectious nature of the disease should be established, e.g., by clinical differential diagnosis in itself or combining this with microbiological data and/or the use of validated questionnaires, depending on the study context and type of infection [40]. Donabedian et al. (2006) concluded that the ten micronutrients which have received a positive opinion for supporting the immune system do not play a supporting role in the treatment of certain ongoing infections [45]. So far, all applications for putative health claims related to stimulation of the immune system and defense against pathogenic microorganisms have been rejected by the NDA Panel. Most of the rejected claims focused on the effects of probiotic bacteria such as *Lactobacillus* and *Bifidobacterium* strains. Claims on probiotics, however, have often been rejected because of a lack of specifying the active ingredient itself, the first step in the scientific assessment of the claim [46,47,48,49]. Other unapproved applications for products claiming immune stimulation and defense against pathogenic microorganisms mainly involved amino acids, antioxidants, oligosaccharides, and fungal compounds, including beta-glucans. The unapproved immune related applications for products containing fungal beta-glucans are listed in Table 2.

These negative evaluations of putative health claims have not stopped research into the effects of specific beta-glucan-containing supplements, such as Yestimun^®^ and pleuran. Since these two supplements are interesting cases of a growing body of evidence, the following sections discuss, in detail, the current state of the evidence in light of the requirements for substantiating health claims.

### 2.3. Yestimun^®^

Yestimun^®^ is an insoluble, highly purified, well-characterized β-glucan from spent brewer’s yeast (*Saccharomyces cerevisiae*) [50]. Brewer’s yeast is grown exclusively on malt and clean spring water and is a natural byproduct of the fermentation process used for beer production. During various possessing steps, these β-glucans are further purified and soluble compounds are removed. This results in a relative β-1,6 glucan side chain binding percentage of 22% with a minimum purity of 85% [50]. Animal studies showed that orally ingested Yestimun^®^ in rats increased the phagocytic activity of granulocytes and monocytes, the percentage of phagocytic cells and nonspecific humoral immune parameters lysozyme, ceruloplasmin and serum γ-globulin [51,52,53]. Phagocytes derived from the β-glucan fed group showed higher respiratory burst and phagocytic activity. When stimulated by LPS, the proliferation rate of lymphocytes was higher in the β-glucan group [53]. Moreover, a study in dogs with inflammatory bowel disease (IBD) showed that animals treated with β-glucan had a decreased level of IL-6 and an increased level of anti-inflammatory IL-10 as compared to untreated control animals [54]. In addition to these animal studies, clinical trials have examined the ability of Yestimun^®^ to increase the body’s defense against invading pathogens. Auinger et al. (2013) performed a placebo-controlled, double-blind, randomized clinical trial in 162 healthy participants with recurring infections and concluded that supplementation with Yestimun^®^ (900 mg/day) for 16 weeks reduced the number of symptomatic common cold infections by 25% compared to placebo (*p* = 0.041) [55]. Another trial with 100 participants confirmed these results by reporting significantly more subjects without a cold episode and its typical symptoms in the β-glucan group compared to the placebo group [56]. Although these clinical studies with Yestimun^®^ showed positive effects on the immune system, the inclusion of these studies into the scientific dossier to support an immune claim was not considered to provide sufficient evidence for substantiating the claim [50,57,58]. The NDA Panel concluded that a cause-and-effect relationship had not been established, mainly because of study design issues: a non-validated questionnaire on common cold was used and limitations of statistical analyses were identified [50].

Recently, another intervention study was conducted that used validated questionnaires on upper respiratory tract infections (URTI) episodes as a primary endpoint [59,60]. Dharsono et al. (2019) concluded that supplementation with Yestimun^®^ reduced the severity of physical URTI symptoms during the first week of an episode, even though the incidence and overall severity of common colds was not shown to be altered in comparison to placebo. Furthermore, accompanying health benefits in terms of lowering blood pressure and improved mood were reported [59]. However, no research has been done on the viral load and the type of viruses/bacteria causing the symptoms. This is an important limitation of the study in substantiating a health claim related to pathogen defense, as the infectious nature of the disease must be established [40]. The nature of the virus might have an impact on incidence, severity, and duration of URTI episodes [59]. In practice, however, it is uncommon to perform routine laboratory tests for the diagnosis of the common cold as it can be caused by many different agents (adenovirus, coronavirus, influenza virus, rhinovirus, etc.) [60,61]. The viral pathogens associated with the common cold may be detected by culture, antigen detection, PCR, or serologic methods. These studies are generally not indicated in patients with the common cold, because a specific etiological diagnosis is only meaningful when considering treatment with an antiviral agent [60,61].

### 2.4. Pleuran

Pleuran is an insoluble polysaccharide (β-(1,3/1,6)-d-glucan), isolated from the fruiting bodies of the edible mushroom *Pleurotus ostreatus*. Pleuran was developed as a dietary supplement to support the immune system and overcome the first signs of exhaustion and fatigue in adults and children [18]. The effect of Imunoglukan P4H^®^, a formulation of pleuran and vitamin C, on respiratory tract infections (RTI) and recurrent respiratory tract infections (RRTI) in children has been investigated in several clinical studies. In an open-label study, a decrease in the frequency of RRTI was observed in 153 children (71.2%). The mean annual incidence of respiratory tract infections in children with a positive response to Imunoglukan P4H^®^ was significantly lower compared to that in unresponsive patients (3.6 vs. 8.9, *p* < 0.001) [62]. Another study examined the effect of Imunoglukan P4H^®^ supplementation on the frequency of RTI in a group of 151 children with RRTI. A comparison between the number and type of RTI during the previous October period was compared to those observed during the intervention period and 6-month follow-up. Supplementation with Imunoglukan P4H^®^ reduced the RRTI rate from 8.88 ± 3.35 episodes in the previous year to 4.27 ± 2.21 episodes in the study year (*p* < 0.001) [63]. Similar efficacy was observed in another prospective open-label study in 194 children in Poland, where a significant decrease in total RTI was reported during the intervention and follow-up periods (4.18 ± 2.132 vs. 8.71 ± 1.89, *p* < 0.001) [64].

However, the design of these studies also shows some weaknesses when considering their usage for substantiating an immune function health claim. Firstly, the prospective open-label design is a weakness. Secondly, the lack of information on viral load and the type of virus/bacteria causing symptoms are limitations. The major weakness in substantiating immune claims for beta-glucans, however, is the fact that Imunoglukan P4H^®^ also contains vitamin C (15% of the recommended dietary allowance). The recommended daily dose contains sufficient vitamin C to make use of the health claim ‘contributes to the normal function of the immune system’. However, when looking specifically at respiratory infections, randomized, placebo-controlled studies have not clearly shown that vitamin C on its own has the potential to prevent them [45,65]. Interestingly, a double-blind, placebo-controlled, randomized trial in children with RRTI examined the effect of the *Pleurotus* extract in itself. Next to self-reporting RRTI symptoms, a validated health questionnaire was used to examine general health status and RRTI symptoms. Vitamin C was used as an ‘active placebo’ to investigate whether the immunomodulatory action, which is clinically manifested in the reduction of RTI, can primarily be attributed to the highly purified *Pleurotus ostreatus* extract [66]. In the *Pleurotus* extract group, 36% of the children did not suffer from any respiratory infections throughout the treatment, compared to 21% in the vitamin C group. *Pleurotus* extract also significantly decreased the frequency of flu and flu-like symptoms, as well as the frequency of lower respiratory tract infections compared to the vitamin C group (0.20 ± 0.55 per 12 months vs. 0.42 ± 0.78 per 12 months, *p* < 0.05). The results of this RCT, that uses validated questionnaires, are promising but need to be confirmed in more studies as multiple intervention studies conducted by independent institutions increase the likelihood of receiving a positive EFSA opinion [20,24]. In addition, with the target population being children with (recurring) respiratory infections instead of the general population, this evidence mainly substantiates an Article 14.1(b) claim. Should such studies yield positive results again, pleuran could apply for an additional health claim, such as the authorized claim for vitamin D discussed in Section 2.2.

Few studies have been conducted with Imunoglukan P4H^®^ in the general population, but several clinical studies have been conducted in other populations susceptible to RTI, e.g., elite athletes [67,68]. Epidemiological evidence suggests that heavy acute or chronic exercise is related to an increased incidence of upper respiratory tract infections in athletes [69]. In a placebo-controlled study, the daily consumption of 100 mg Imunoglukan P4H^®^ for two months prevented post-exercise immune suppression in elite athletes, and in another study, supplementation reduced RTIs (even though this was measured by a non-validated questionnaire) [67,68]. These studies demonstrate the potential for the use of Imunoglukan P4H^®^ as an immunostimulant in elite athletes. Still, follow-up studies will have to be conducted following the guidance documents to qualify as supportive evidence for a possible health claim. When such studies would again yield positive results, Imunoglukan P4H^®^ could apply for an additional health claim, such as the authorized claim for vitamin C: ‘Vitamin C contributes to the maintenance of the normal function of the immune system during and after intense physical exercise’.

## 3. Trained Immunity

When smallpox vaccination was introduced about 200 years ago and up to its discontinuation in 1980, positive side effects were noted by physicians, such as protection against measles, scarlet fever, and whooping cough [70]. Investigation of these observations led to evidence in 1956 of the ‘ring zone phenomenon’, i.e., the production of soluble antivirals in infected chicken embryos and cell cultures. With the help of modern immunological and bioengineering methods, it was later possible to demonstrate that these effects are based on the activation of lymphoreticular cells and the regulatory effect of certain cytokines within the context of the nonspecific immune system [70]. In recent years, it has become evident that cells of the innate immunity may be primed upon encounter with certain pathogens or molecular patterns associated to pathogens, thereby acquiring a higher resistance to a second infection against the same or unrelated pathogens [71,72,73]. This concept, known as ‘trained immunity’, gives rise to the development of trained immunity-based vaccines (TIbV), defined as vaccine formulations that induce training in innate immune cells, and the use of nonspecific immunostimulants as ‘trained immunity’ inducers [71,73]. TIbV could be used during viral outbreaks to confer nonspecific protection as well as to enhance adaptive specific immune responses. Moreover, the ability of TIbV to promote responses beyond their nominal antigens may be useful when conventional vaccines are not available or when multiple co-infections and/or recurrent infections occur in susceptible individuals [71]. Several drugs used in cancer therapy (e.g., BCG-vaccine, mifamurtide) have recently been shown to be able to elicit an enhanced immune response in monocytes after nonspecific restimulation [74]. Interestingly, in vitro and animal studies showed that fungal beta-glucans are also able to elicit trained immunity through activation of the pattern recognition receptor (PRR) dectin-1 [71,73,75,76,77]. It has been speculated that the increased expression of certain PRRs in innate trained cells, as well as the release of typical innate immunity cytokines, such as IL-1β, contribute to enhance adaptive T-cell responses [71]. The pharmaceutical application of beta-glucans has been carried out within TCM for decades, but until now, this concept is largely unknown in Western medicine. The new insights and developments in trained immunity may lead to the possible application of these drugs and fungal beta-glucans as NSIs and/or adjuvants in TIbV in Western medicine. The following sections discuss, in detail, the currently available evidence from clinical intervention studies in humans investigating nonspecific immunity. Subsequently, the pharmaceutical application of fungal glucans in Western medicine and the substantiation of the required evidence for this are discussed.

### 3.1. BCG Vaccine

The bacillus Calmette–Guérin (BCG) vaccine contains live-attenuated *Mycobacterium bovis* bacilli that protect against tuberculosis, one of the world’s deadliest infectious diseases [78]. In countries where tuberculosis is common, one dose is recommended in healthy babies as close to birth as possible. In addition, it is sometimes used in cancer therapy, such as in the treatment of bladder cancer [79,80]. For example, the BCG vaccine is indicated as curative treatment of carcinoma in situ of the urothelium of the bladder and as an adjuvant after transurethral resection of a primary or recurrent superficial papillary carcinoma of the urothelium of the bladder. Upon instillation, the BCG vaccine locally stimulates the immune system (with an increase in granulocytes, monocytes/macrophages and T-lymphocytes). Early in vitro and animal studies indicated that BCG vaccine immunization could induce nonspecific, protective effects against other pathogens. For example, it was observed that mice vaccinated against tuberculosis were also found to be protected when secondary infections with *Staphylococcus aureus*, *Listeria monocytogenes*, *Salmonella typhimurium*, or *Schistosoma mansoni* had occurred [76,81,82,83,84]. Only in recent years, attempts have been made to identify mechanistic events responsible for the nonspecific effects in humans. These efforts resulted in the identification of a plethora of methods to identify human biomarkers, from cytokine responses to epigenetic changes [85]. A Dutch study on a small group of BCG-vaccinated young adults demonstrated that non-mycobacterial stimuli were able to induce heterologous Th1 and Th17 cytokines, such as IFN-γ and IL- 17, up to 1 year after receiving the vaccine. The vaccination only induced a primed status of the cells to respond more strongly to secondary microbial stimulation. Without stimulation, no higher production of these cytokines was seen [86]. The beneficial effects of BCG vaccination are therefore suggested to be the induction of the innate immunity reprogramming expressed as the long-term sustained changes in the nonspecific resistance against infections [76]. As described in Table 3, following the identification of these beneficial effects, more clinical studies on BCG-induced trained immunity have been conducted in healthy subjects, neonates, infants, as well as the elderly [86,87,88,89,90,91,92,93].

Trained immunity responses induced by BCG vaccination were shown to be dependent on the engagement of the intracellular receptor nucleotide-binding oligomerization domain 2 (NOD2) by the peptidoglycan component muramyl dipeptide (MDP) [84,96]. MDP is the smallest naturally occurring nonspecific immunostimulating component of the cell wall of *Mycobacterium*. MDP is derived from various sources and is present in human peripheral blood [97,98].

### 3.2. Mifamurtide

Mifamurtide (muramyl tripeptide phosphatidylethanolamine, MTP-PE) is a fully synthetic derivative of MDP. Mifamurtide has the same immunostimulating effect as natural MDP. As mifamurtide is lipophilic, it may be incorporated into liposomal lipid bilayers and subsequently phagocytosed by macrophages and monocytes. These activated cells may then selectively target and destroy tumor cells without affecting normal cells [99]. Both mifamurtide and MDP stimulate immune responses by binding to nucleotide-binding oligomerization domain-containing protein 2 (NOD2), an intracellular pattern-recognition receptor molecule expressed mainly in monocytes, macrophages, and dendritic cells [99]. Mutations in NOD2 are frequently observed in patients with Crohn’s disease, an autoimmune disorder, suggesting the significance of the MDP–NOD2 pathway in activating immunity. As more became known about the MDP–NOD2 pathway, structural modifications of MDP and its derivatives have been extensively studied in an attempt to increase adjuvant activity and boost the immune response effectively for clinical use in the treatment of cancer and other diseases [100]. By binding to NOD2, mifamurtide activates the NF-κB pathway that leads to an increased production of proinflammatory cytokines such as TNF-α, IL-1, IL-6, IL-8, interferon gamma (IFN-gamma), and increased levels of immune stimulation markers plasma neopterin and serum C-reactive protein [99,100]. There are two formulations of mifamurtide, the free-drug form (MTP-PE) and the liposomal-encapsulated form (L-MTP-PE). The maximum tolerated dose (MTD) of mifamurtide is 6 mg/m^2^, with a moderate toxicity that has some dose-limiting side effects such as chills, fever, malaise, and nausea. The precise mechanism of action of cell activation, immunomodulation, and tumor reduction by mifamurtide in humans is unknown [99]. A recent study by Mourits et al. (2020) investigated whether variation in circulating levels of MDP can modulate trained immune responses induced by BCG vaccination in vivo and explain the variability of response between individuals. They concluded that circulating pre-vaccination MDP concentrations correlated with systemic inflammation and induction of trained immunity after BCG vaccination, but not with specific T-cell cytokine responses. In addition, BCG vaccination was shown to result in a sustained increase in circulating levels of MDP, but this change in MDP did not affect trained immune responses or specific memory immune responses [96]. As described in Table 3, a pharmacokinetic and pharmacodynamic study in healthy adults showed an increase of serum concentrations of IL-6, TNF-α, and CRP after infusion of L-MTP-PE **[94]**. However, in this study, no restimulation with a second infection was investigated to see the effects of mifamurtide on trained immunity. For follow-up studies, it would be interesting to further unravel our understanding of trained immunity by conducting clinical trials with MDP or mifamurtide and restimulating with a second infection, as previously done with BCG.

### 3.3. Fungal Glucans and Trained Immunity

In addition to the above-described approved medicinal products that induce trained immunity, in vitro and animal studies show that beta-glucans from ‘medicinal’ mushrooms may also play an interesting role in this new field of immunology. For example, initial stimulation of myeloid cells by fungal β-glucan has been shown to promote control of subsequent infection with bacterial pathogens [73,75,76,77]. Furthermore, as beta-glucans are inexpensive and well-tolerated compared to most pharmaceutical products, and can be taken orally, beta-glucan appears to be a promising candidate to enhance the immune response [101]. Compared to the BCG vaccine, hardly any RCTs have been performed to study putative immunostimulating effects of orally administered beta-glucans in humans. In a randomized open-label intervention pilot study, the potential immunostimulating effects of commercially available orally administered water-insoluble beta-glucan in healthy participants were investigated [95]. This supplement, Glucan 300^®^, derived from baker’s yeast, was found to be the most active compared to other beta-glucans in a mouse study and its use was considered safe for human consumption [73]. Leentjens et al. (2014) concluded that beta-glucan was barely detectable in the serum of volunteers at all studied time points. In addition, neither the production of cytokines nor the microbicidal activity of leukocytes was affected by orally administered beta-glucan, as described in Table 3 [95]. This lack of reported effects may be attributed to the selected commercial dietary supplement, which may have degraded the beta-glucans before absorption could occur. A highly purified pharmaceutical composition optimized for oral administration may have immunostimulating effects in humans. It was further recommended that intravenous administration of a pharmaceutical preparation may exert immunostimulating effects [95]. The most studied intravenously administered beta-glucan, lentinan, is clinically approved in several countries in Asia [102].

#### 3.3.1. Lentinan

Lentinan is a polysaccharide isolated from the fruiting body of the shiitake mushroom and has been used in Asia for thousands of years to improve health [102]. Lentinan is approved for treating multiple types of cancer, hepatitis, and other diseases in China and as an adjuvant for stomach cancer therapy in Japan [102]. Intravenous injection of lentinan is clinically approved with an average dose of 1–1.5 mg/day. Furthermore, lentinan is available in capsules and tablets and taken orally as a traditional medicine. The primary structure of β-glucan in lentinan is composed of a β-(1–3)-glucose backbone with two (1–6)-β-glucose branches of every five glucose units [17]. Recently, a systematic review including 38 RCTs (3117 patients), investigated the clinical effectiveness of intravenously administered lentinan as an adjuvant therapeutic drug in the treatment of patients with lung cancer [102]. It was concluded that lentinan had a favorable safety profile compared to chemotherapy, hormone therapy and immunotherapy. In addition, lentinan was considered effective not only for improving quality of life, but also for promoting the efficacy of chemotherapy in the treatment of lung cancer [102]. Early in vitro and animal studies indicated that lentinan could induce nonspecific, protective effects against pathogens [103,104,105,106]. Moreover, clinical data from TCM indicate that lentinan is a nonspecific immunostimulant and a biological response modifier with proven efficacy in treating viral infections, such as hepatitis and HIV [107,108,109]. In view of these results, it appears that lentinan is a promising candidate for follow-up studies into trained immunity in healthy subjects, as previously performed with BCG. However, the question remains how this routine Eastern practice can be translated into Western medicine or novel food [110,111]. For novel foods, the scientific dossier must provide evidence that no adverse effects are elicited by consuming the product and consequently, kinetics, toxicology, nutritional information and allergenicity must be analyzed [112]. Information from nonclinical studies required by Western standards of evidence is often lacking for these medicinal mushrooms. During the early preclinical development process, a drug candidate must go through several steps, such as determination of bioavailability, pharmacokinetics, pharmacodynamics, absorption, distribution, metabolism, and elimination (ADME), and preliminary studies aimed at investigating the candidate’s safety including genotoxicity, mutagenicity, safety pharmacology, and general toxicology [113]. Although side effects of lentinan are rare, anaphylaxis after intravenous administration has been reported in some clinical cases [114]. Anaphylaxis could be caused by a toxic reaction to lentinan as there are also case reports of contact dermatitis, asthma, rhinitis, and hypersensitivity pneumonitis in shiitake workers [115]. These cases indicate the need to conduct additional (preclinical) studies, as regulators take the safety of novel food and new agents as their primary consideration [112]. In addition to safety studies, another issue for purified or crude medicinal mushroom extracts is ensuring good manufacturing practice (GMP) standards with batch-to-batch consistency, end-product stability, and consistent analytical profiles of products [116,117]. In the absence of suitable methods for standardization and characterization of fungal glucans, guaranteeing GMP is particularly difficult. The potential molecular mechanisms underlying the induction of innate immune memory by medicinal mushrooms is a complex interaction between immunological, metabolic, and epigenetic changes through many as yet unknown pathways [102]. Bringing traditional Eastern practices into Western practice does not require a full understanding of the mechanism of action, but requires clinical studies that provide data that the clinical community will accept [110]. An example of a pharmaceutical beta-glucan preparation currently in clinical development for cancer according to Western regulations is Imprime PGG.

#### 3.3.2. Imprime PGG

Imprime PGG is an intravenous formulation of a yeast-derived, uncharged, water-soluble, 1,3–1,6 beta glucan purified from the cell wall of a proprietary, non-recombinant, strain of *Saccharomyces cerevisiae*. Initial in vitro and animal studies provided insights into bioavailability, pharmacokinetics, pharmacodynamics, ADME, genotoxicity, mutagenicity, safety pharmacology, and general toxicology, information necessary for initiating later clinical trials [116,117,118,119,120,121,122,123,124,125,126,127,128,129,130,131,132]. Moreover, ex vivo human whole blood studies demonstrated that Imprime-induced responses were consistent with the innate immune activation elicited by a pathogen. Imprime binding and functional activation of these innate effector cells was critically dependent on Imprime first forming an immune complex with endogenous IgG anti-beta-glucan Ab (ABA). The formation of Imprime–ABA complexes induced significant activation of complement proteins as well as phenotypic activation and chemokine production by innate immune cells [133]. As a follow-up to these studies, two randomized, double-blind, placebo-controlled phase 1 dose escalation studies evaluating Imprime in healthy participants were conducted [134]. Recently, Bose et al. (2019) conducted a phase 1 study in healthy participants, examining the relationship between ABA levels and Imprime-mediated innate immune activation. Consistent with ex vivo results, Imprime–ABA complexes induced significant activation of complement proteins as well as phenotypic activation and chemokine production by innate immune cells, as described in Table 3 [135]. These results demonstrate that intravenously administered beta-glucan has immunostimulating effects in vivo and thus may be of significance in trained immunity. Based on the favorable safety and tolerability results observed in these studies, phase 2 studies in which Imprime was administered in combination with antitumor monoclonal antibodies have been initiated in cancer patients [136,137].

## 4. Conclusions

This review has highlighted the potential application of fungal beta-glucans—immunomodulators that have been used in traditional Chinese medicine for centuries—in nutrition and medicine. From this review, it can be concluded that fungal glucans may play a promising role within both perspectives, and that there are possibilities to give traditional medicine an immunological application in both medicinal products and foods. Depending on the dosage, formulation, efficacy, safety profile, and route of administration, the immunomodulating effects that can be expected from fungal beta-glucans can either be considered a pharmaceutical effect (treating or curing a disease) or as a health effect originating from foods, focusing on the prevention of negative health effects.

In Europe, claims on health benefits are strictly regulated, with EFSA reviewing the scientific evidence that supports putative statements about health effects. As shown in this paper, all applications for putative health claims related to stimulation of the immune system and defense against pathogenic microorganisms have so far been rejected. Since EFSA has only approved immune claims for six vitamins and four essential trace elements, it can only be speculated that the temptation to add these ingredients to products is growing, rather than stimulating research into innovative foods. Comparing the evidence base of the putative health effects of fungal beta-glucan supplements with the guidance documents on immune support health claims, but even more importantly, the guidance documents on substantiating immune stimulation and pathogen defense by food products, it is shown that fungal glucans could play a role in supporting and maintaining health and, thus, can be seen as a good health-promoting substance from food—which could mean that this effect may also be claimed if approved.

In addition to these developments related to food uses of beta-glucan-containing supplements, beta-glucans could also hold a novel position in Western medicine, as the concept of trained immunity is relatively new and has not been investigated to a larger extent. The new insights and developments in trained immunity may lead to the possible application of fungal beta-glucans as NSIs in Western medicine. Due to the experience from Asian medicine and the relatively favorable safety profile, lentinan (i.v.) could potentially be a suitable fungal glucan within this new field of immunity. However, additional (preclinical) safety studies must first be performed to be eligible as a medicine in Europe. Imprime PGG, which is currently going through the stages of drug development, is another fungal beta-glucan worth investigating. Finally, given the different ways to purify and process beta-glucans, one of the biggest challenges remains the standardization and proper characterization of the active compounds themselves. However, with the help of modern immunological and biotechnological methods, increasing insights are gained into immunomodulating fungal beta-glucans, with potential applications both in foods and pharmaceutical products.

## Figures and Tables

**Figure 1 nutrients-13-01333-f001:**
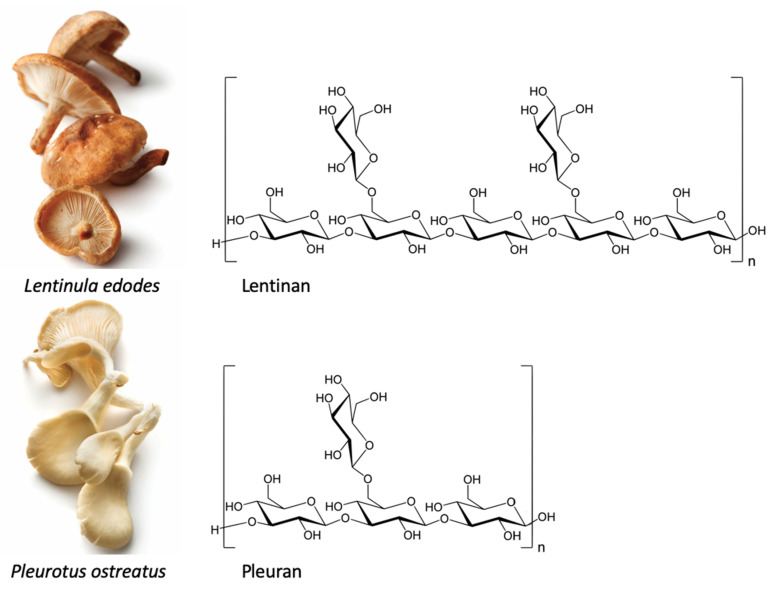
Shiitake (*Lentinula edodes*) and pearl oyster mushroom (*Pleurotus ostreatus*) and their specific beta-glucan structures.

**Table 1 nutrients-13-01333-t001:** Changes in biomarkers that are proposed as outcomes for substantiating health claims related to immune function [31].

Described Changes	Examples
Immune biomarkers	- Numbers of various lymphoid subpopulations in the circulation - Proliferative responses of lymphocytes - Phagocytic activity of phagocytes - Lytic activity of NK cells and cytolytic T cells - Production of cellular mediators - Serum and secretory immunoglobulin levels - Delayed-type hypersensitivity responses
Inflammation biomarkers	- C-reactive protein (CRP) - Interleukins (e.g., IL-6, IL-8, IL-10) - Tumor necrosis factor-α (TNF-α)
Short-chain fatty acid production in the gut	- Acetate - Propionate - Butyrate
Structure of the intestinal epithelium	- Composition of cells (e.g., enterocytes, Paneth cells, M cells)
Composition of gut microbiota	- Phyla (e.g., Actinobacteria, Firmicutes, Bacteroidetes, Proteobacteria)

**Table 2 nutrients-13-01333-t002:** Claims regarding fungal beta-glucan applications related to the immune system.

Claim Type	Nutrient, Substance, Food or Food Category	Claim	Non-Authorization/Discontinuation Based on Criteria	Health Relationship	EFSA Opinion/Journal Reference	Entry ID
Art. 13(1)	Beta-glucan (WGP)	For immunity. Strengthens immunity.	(2) Rejected on the basis of an unclear health relationship or no clear association with health.	Immune system	2011;9(6):2228	1792
Art. 13(1)	Beta-glucan + olive leaf extract	Supports the body’s own defense mechanism/immunity. Maintains natural defense mechanism/immunity. Helps strengthen natural immunity.	(2) Rejected on the basis of an unclear health relationship or no clear association with health.	Immune function/immune system	2011;9(4):2061	1793
Art. 13(1)	Beta-glucan of *Saccharomyces cerevisiae*	Beta-glucan from yeast as immunomodulators. Beta-glucan from yeast support of natural defenses.	(2) Rejected on the basis of an unclear health relationship or no clear association with health.	Immune system	2011;9(6):2228	847
Art. 13(1)	Beta-glucan of *Saccharomyces cerevisiae*	Beta-glucan from yeast as immunomodulators. Beta-glucan from yeast support of natural defenses.	(2) Rejected on the basis of an unclear health relationship or no clear association with health.	Increasing nonspecific serum IgA secretion	2011;9(6):2228	1944
Art. 13(1)	WGP beta-glucan; (WGP^®^ (1,3)-b-d-glucan); (from *Saccharomyces cerevisiae*)	WGP beta-glucan contributes to the normal function to the immune system. WGP beta-glucan naturally contributes to adequate immune responses. The daily dietary supplementation with WGP beta-glucan promotes the normal function of the immune system. WGP beta-glucan enhances the production and activity of macrophages and neutrophils. Thus, it plays an important role in the adequate function of the immune system. WGP beta-glucan contributes to maintain the normal function of the upper respiratory tract.	(3) Rejected on the basis of an unproven cause and effect relationship: no evidence (yet) for a relationship between intake and effect.	Maintenance of the upper respiratory tract defense against pathogens by maintaining immune defenses.	2011;9(6):2248	1910
Art. 13(5)	Yestimun^®^	Daily administration of Yestimun^®^ helps to maintain the body’s defense against pathogens.	(3) Rejected on the basis of an unproven cause and effect relationship: no evidence (yet) for a relationship between intake and effect.	N/A	Q-2012-00761Commission Regulation (EU) No 1154/2014 of 29/10/2014	N/A
Art. 13(5)	Yestimun^®^, consisting of (1,3)-(1,6)-β-d-glucans of brewer’s yeast cell wall (100% *Saccharomyces cerevisiae*)	Daily administration of Yestimun^®^ strengthens the body’s defense during the cold season.	(3) Rejected on the basis of an unproven cause and effect relationship: no evidence (yet) for a relationship between intake and effect.	N/A	Q-2008-667Commission Regulation (EU) 432/2011 of 04/05/2011	N/A
Art. 13(1)	*Lentinula edodes* (common name: Shiitake)	Contributes to natural immunological defenses.	(3) Rejected on the basis of an unproven cause and effect relationship: no evidence (yet) for a relationship between intake and effect.	Immune function/immune system	2011;9(4):2061	3774
Art. 13(1)	*Lentinula edodes* (common name: shiitake)	Contributes to natural immunological defenses.	(2) Rejected on the basis of an unclear health relationship or no clear association with health.	Stimulation of immunological responses	2011;9(4):2061	2075
Art. 13(1)	*Pleurotus ostreatus* (oyster mushroom)	Contributes to natural immunological defenses.	(2) Rejected on the basis of an unclear health relationship or no clear association with health.	Immune function/immune system	2011;9(4):2061	3521
Art. 13(1)	Brewer’s yeast	Strengthens immunity	(2) Rejected on the basis of an unclear health relationship or no clear association with health.	Immune function/immune system	2010;8(10):1799	1384
Art. 13(5)	Immune balance drink, containing vitamin C, green tea, grape skin, grape seed, and shiitake mushroom extract	The immune balance drink activates body’s defense.	(3) Rejected on the basis of an unproven cause and effect relationship: no evidence (yet) for a relationship between intake and effect.	N/A	Q-2009-517Commission Regulation (EU) No 958/2010 of 22/10/2010	N/A
Art. 13(1)	Active hexose correlated compound (AHCC)	Activates immune system, exert potential effects on the immune system—stimulating immunity.	(2) Rejected on the basis of an unclear health relationship or no clear association with health.	Stimulation of immunological responses	2011;9(4):2061	3139
Art. 13(1)	Herbal yeast plasmolysate (*Saccharomyces cerevisiae*)	Strengthens the body’s defense system. Increases immunity.	(2) Rejected on the basis of an unclear health relationship or no clear association with health.	Immune function/immune system	2011;9(4):2061	1817

**Table 3 nutrients-13-01333-t003:** Clinical trials on trained immunity.

Population	Intervention	Conclusions	Ref.
20 Healthy individuals (age: 20–36 years)	Participants would receive a BCG vaccination from the public health agency for traveling to or working in countries where tuberculosis is prevalent.	The production of TNF-α and IL-1β to mycobacteria or unrelated pathogens was higher after 2 weeks and 3 months post-vaccination, but these effects were less pronounced 1 year after vaccination. However, monocytes recovered 1 year after vaccination had an increased expression of pattern recognition receptors such as CD14, toll-like receptor 4 (TLR4) and mannose receptor, and this correlated with an increase in proinflammatory cytokine production after stimulation with the TLR4 ligand lipopolysaccharide.	[86]
30 Healthy Dutch male participants (age: 19–37 years)	Participants received either BCG (*n* = 15) or placebo (the diluent used to dissolve BCG) (*n* = 15). One month after placebo or BCG vaccination, all volunteers received a single dose of yellow fever vaccine.	BCG-vaccinated volunteers displayed a significant reduction of viremia compared to the placebo group, which highly correlated with enhanced IL-1β production.	[87]
20 Healthy, BCG-naive volunteers (age: 18–35 years)	Ten subjects received standard dose (0.1 mL of the reconstituted vaccine) of intradermal BCG vaccination 5 weeks prior to challenge infection. Ten controls received no vaccination. Five weeks after BCG vaccination, both groups were exposed to bites of five *Plasmodium falciparum* NF54 strain infected *Anopheles stephensi* mosquitoes (sporozoite challenge).	BCG vaccination altered some of the clinical, immunological, and parasitological outcomes of malaria infection in a subset of volunteers. Earlier NK cell and monocyte activation in this subset of vaccinated volunteers is consistent with the possibility that induction of trained innate immunity in vivo may have functional activity against a heterologous pathogen in humans.	[88]
212 Neonates; BCG vaccinated (*n* = 119) BCG naïve (*n* = 93)	Participants were randomized 1:1 to undergo vaccination with BCG (0.05 mL) intradermally within 10 days of birth or to receive no BCG vaccine.	BCG-vaccinated infants had increased production of IL-6 in unstimulated samples and decreased production of interleukin 1 receptor antagonist, IL-6, and IL-10 and the chemokines macrophage inflammatory protein 1α (MIP-1α), MIP-1β, and monocyte chemoattractant protein 1 (MCP-1) following stimulation with peptidoglycan (TLR2) and R848 (TLR7/8). BCG-vaccinated infants also had decreased MCP-1 responses following stimulation with heterologous pathogens.	[89]
40 Healthy volunteers (age: 20–25 years)	Participants received either live attenuated BCG vaccine (*n* = 20) or placebo (*n* = 20), followed by intramuscular injection of trivalent influenza vaccine 14 days later.	In BCG-vaccinated subjects, HI antibody responses against the 2009 pandemic influenza A(H1N1) vaccine strain were significantly enhanced compared with the placebo group. Additionally, apart from enhanced proinflammatory leukocyte responses following BCG vaccination, nonspecific effects of influenza vaccination were also observed, with modulation of cytokine responses against unrelated pathogens.	[90]
15 Healthy individuals (age: 20–34 years)	Participants received inactivated gamma-irradiated BCG (γBCG). The inactivated BCG was cultured for 6 weeks to confirm inactivation.	γBCG vaccination in volunteers had only minimal effects on innate immunity. The results indicate that γBCG induces long-term training of innate immunity in vitro. In vivo, γBCG induces effects on innate cytokine production are limited.	[91]
198 Elderly patients (age >65 years)	Participants received BCG (*n* = 100) or placebo (*n* = 98) vaccine and were followed for 12 months for new infections.	At interim analysis, BCG vaccination significantly increased the time to first infection. The incidence of new infections was 42.3% after placebo vaccination and 25.0% after BCG vaccination; most of the protection was against respiratory tract infections of probable viral origin.	[92]
158 Infants	Infants received BCG within 7 days of birth (*n* = 80). Controls (*n* = 78) were bled 4 days post-randomization, and at age 3 and 13 months.	BCG vaccination of Danish newborns did not induce nonspecific in vitro cytokine responses.	[93]
21 Healthy participants(age: 21–59 years)	Participants received a single 4 mg i.v. infusion of L-MTP-PE over 30 min.	Serum concentrations of IL-6, TNF-α, and CRP increased following L-MTP-PE infusion. Maximum observed increases in IL-6 and TNF-α occurred at 4 and 2 h, respectively, returning toward baseline by 8 h post-dose.	[94]
15 Healthy male participants(age: 19–24 years)	Beta-glucan (*n* = 10) or the control group (*n* = 5). Subjects in the glucan group ingested beta-glucan 1000 mg once daily for 7 days. Water-insoluble beta-glucan derived from baker’s yeast (*S. cerevisiae*) sold as a dietary supplement (Glucan #300, BG). This preparation has a purity of at least 83% guaranteed by the manufacturer.	Beta-glucan was barely detectable in serum of volunteers at all time points. Neither cytokine production nor microbicidal activity of leukocytes were affected by orally administered beta-glucan.	[95]

## Data Availability

Not applicable.

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
