# Peer review of "Immunomodulating Effects of Fungal Beta-Glucans: From Traditional Use to Medicine"

_nutrients, 2021, doi:10.3390/nu13041333_

Round 1

Reviewer 1 Report

Some Acronyms require definition

Line 147 The European Commission subsequently asks EFSA , Please define EFSA

Line 395 pplication of beta-glucans has been carried out within TCM, Please define TCM

Reviewer 2 Report

The immunomodulatory effects of fungal beta-glucan is a well issue in recent decades. However, there are a number of problems with the manuscript as presented.

  1. There are many immune biomarkers from the health claims in the table 1, the authors should introduce the overall responses of immune system in the p1 INTRODUCTION section, such as T cells.
  2. Fungal beta-glucans are bioactive molecules in the immune system, and the specific receptors also associated with the structure of fungal glucans, the authors should descript the biological mechanisms of the immunomodulatory actions.
  3. Dietary beta-glucan can be uptake by GI cells?
  4. P5-6, line 180-227, the EU health claims had been inducted in this article, the paragraph is very long to descript the effects of vit C and vit D on immune system, but not focus on the role of the dietary beta-glucan.
  5. In the trained immunity section, the authors should descript the role of beta-glucan in the vaccine. There are many studies had been reported, also included the review article.
  6. There are many fungi have been used in Traditional Chinese Medicine for centuries, and also used in the food supplementation. But the authors didn’t discuss about the glucan of TCM in this article.

Round 2

Reviewer 2 Report

The manuscript has been significantly improved.